# Preference and retention of daily and event-driven pre-exposure prophylaxis for HIV prevention: a prospective cohort in Can Tho city, Viet Nam

Van Thi Thuy Nguyen [ID],[1] Vu Quoc Dat [ID],[2,3] Huynh Minh Truc,[4]
Pham Nguyen Anh Thu,[4] Doan Thi Thuy Linh,[5] Cheryl Johnson,[6]
Rachel Clare Baggaley,[7] Huong Thi Thu Phan[5]

[1]WHO Viet Nam, World Health Organization, Geneva, Switzerland
[2]Department of Infectious Diseases, Hanoi Medical University, Hanoi, Viet Nam
[3]Hanoi Medical University Hospital, Hanoi Medical University, Hanoi, Viet Nam
[4]City Center for Disease Control and Prevention, Can Tho, Viet Nam
[5]Viet Nam Administration for HIV/AIDS Control, Hanoi, Viet Nam
[6]Department of HIV/AIDS, World Health Organization, Geneve, Switzerland
[7]World Health Organization, Geneva, Switzerland

**Correspondence to**
Dr Vu Quoc Dat;
quocdat181@yahoo.com

## ABSTRACT

**Objective** Pre-exposure prophylaxis (PrEP) was introduced in Viet Nam in 2017, but data on oral PrEP preference and effective use beyond 3 months are limited. We aimed to evaluate PrEP preferences for PrEP, factors influencing uptake, choice and effective use, as well as barriers to PrEP.

**Methods** This is a prospective cohort study in Can Tho, Viet Nam. Participants who were eligible for PrEP and provided informed consent were interviewed at baseline on demographic information, willingness to pay, reasons for choosing their PrEP regimen and the anticipated difficulties in taking PrEP and followed up at 3 months, 6 months and 12 months after PrEP initiation.

**Findings** Between May 2020 and April 2021, 926 individuals at substantial risk for HIV initiated PrEP. Of whom 673 (72.7%) choose daily PrEP and 253 (27.3%) choose event-driven (ED)-PrEP. The majority of participants were men (92.7%) and only 6.8% were women and 0.5% were transgender women. Median participant age was 24 years (IQR 20–28) and 84.7% reported as exclusively same-sex relationship. The three most common reasons for choosing daily PrEP were effectiveness (24.3%) and unplanning for sex (22.9%). Those opting for ED-PrEP also cited effectiveness (22.7%), as well as convenience (18.0%) and easier effective use (12.0%). Only 7.8% of PrEP users indicated they were unwilling to pay for PrEP and 76.4% would be willing to pay if PrEP were less than US$15 per month. The proportion of user effectively using PrEP at 12 months was 43.1% and 99.2% in daily PrEP and ED-PrEP users, respectively.

**Conclusions** ED-PrEP was preferred by more than a quarter of 23.5% of the participants and there was little concern about potential adverse events. High rates of effective use were reported by ED-PrEP users. Future research to inform implementation of PrEP in Viet Nam is needed to develop ways of measuring adherence to ED-PrEP more accurately and to understand and address difficulties in taking daily PrEP use.

## INTRODUCTION

As of December 2020, Viet Nam reported that there were 215 220 people with HIV, in which there were 12 200 new HIV infections

## STRENGTHS AND LIMITATIONS OF THIS STUDY

⇒ We conducted the first study on preference, retention and factors associated with these in pre-exposure prophylaxis (PrEP) use in Viet Nam.
⇒ The major limitations related to the study design of a single centre.
⇒ The self-statement of adherence among event-driven PrEP users in this study could contribute to overestimate of the retention.

and 1681 AIDS-related deaths in 2020.[1] The HIV epidemic in Viet Nam is concentrated in key populations including people who inject drugs (PWID), men who have sex with men (MSM) and female sex workers (FSWs). It is estimated that there are approximately 200 000 MSM in Viet Nam.[2] In recent years, HIV prevalence has increased in the MSM population, from 5.1% in 2015 to 13.3% in 2020, while prevalence was stable in PWID populations (12.7% in 2019) and FSWs (3.1% in 2020).[1]

Since 2016, the WHO has recommended oral pre-exposure prophylaxis (PrEP) to further reduce new infections among populations where HIV incidence and risk is high. Following this guidance, between June and December 2017, Viet Nam updated their national guidelines and started initial PrEP implementation in Hanoi and Ho Chi Minh city. Since then, PrEP implementation in Viet Nam has continued to expand and as of August of 2021, there were nearly 32 000 persons using PrEP across 200 clinics in nearly half of all provinces in the country.[3] Current national guidelines recommend daily PrEP (tenofovir disoproxil fumarate (TDF) coformulated with emtricitabine (FTC) or lamivudine (3TC)) for populations at substantial risk and event-driven PrEP (ED-PrEP) (TDF/

XTC) for MSM who have less frequent sex (<2 times per week).[4]

While oral PrEP continues to expand and be an effective option for many, recent evidence has highlighted that more differentiated service delivery options are needed. In particular, ED-PrEP provides an effective option which removes the need for daily doses and for use before and after high risk sex. Among MSM, ED-PrEP has been shown to reduce HIV transmission by up to 86%.[5] Studies have also shown that MSM may often prefer ED-PrEP over daily oral PrEP because of its convenience. In a US survey, 74.3% of MSM who were hesitant to start oral daily PrEP indicated that they would be more willing to try oral ED-PrEP.[6] In Thailand, some PrEP users considered daily regimens the easiest to use, as it could be incorporated into daily routines and did not require planning for sex. These men expressed concerns, however, about the long-term safety and affordability of daily oral dosing.[7] Study participants appreciated oral ED-PrEP for minimising drug exposure and potential adverse events. They considered ED-PrEP an attractive choice for MSM who had infrequent sex, were able to plan for sex and had the ability to take the postsex dose.[7]

Despite the potential benefits of ED-PrEP, it is little known about preference and uptake of ED-PrEP among MSM in Viet Nam. Thus, this study aims to assess both preferences as well as actual uptake and continuation of oral daily and ED-PrEP among MSM in Viet Nam to inform future programming. In addition, difficulties related to PrEP uptake and continuation including COVID-19-related issues were explored to inform future differentiated PrEP service delivery models.

## METHODS
### Study design and participants
We conducted a prospective study in all 11 PrEP clinics in Can Tho which has the highest HIV prevalence among MSM (22.7%).[2] MSM were referred to the PrEP clinics from community-based HIV testing led by MSM groups or via self-referral. All clinics were integrated with HIV testing and/or ART services. PrEP eligibility was evaluated following the national guideline: (1) confirmed HIV-negative status, (2) no signs and symptoms of acute HIV infection and (3) at substantial risk for HIV infection within past 6 months. We defined substantial risk as any of the following: individual engaged in condomless anal or vaginal sex, having at least two sexual partners, reported sexual partner with substantial risk for HIV infection or having a sexual partner with HIV but not currently on ART or with unknown/detectable viral load (>200 copies/mL), who had been previously diagnosed with a sexually transmitted infection (STI), and who reported having multiple courses of PEP and continued sexual risk behaviour. Only eligible participants aged 16 years and over who agreed to participate and provide written informed consent were recruited for the study.

### Study procedure and data collection
In the community-based setting, PrEP screening and offering different PrEP regimen is part of HIV post-test counselling.[8] Clients who were interested in PrEP will be referred to a PrEP clinic. At the PrEP clinics, clients were evaluated based on their behavioural risk to assess PrEP eligibility. ED-PrEP was offered for MSM who have infrequent sex (≤2 times per week on average) and is usually able to plan for sex at least 2 hours in advance, or who can delay sex for at least 2 hours or their own preference of ED-PrEP. During the screening, the clinic staff explained what PrEP is, the benefits and the differences between daily PrEP and ED-PrEP and let the client decide. After the clients chose their preferred PrEP regimen, they were invited to provide informed consent and participate in the study and provide written informed consents. Daily PrEP regimens were offered based on the availability of the antiretroviral including TDF/FTC or TDF/3TC or TDF. ED-PrEP regimens were offered as TDF/FTC or TDF/3TC.

National guidelines recommend follow-up with all PrEP clients at health facilities starting with 1–2 months after PrEP initiation and then quarterly thereafter. We used a questionnaire consisting of six questions on willingness to pay (closed-end questions. Willingness-to-pay estimates were reported in 2000 Vietnam Dong (VND) during the interview and converted to US dollar (2021) for this analysis (US$1=VND23 529). Semistructured interviews were used to understand potential barriers to PrEP use. PrEP users were monitored following Viet Nam Ministry of Health's guidelines including HIV testing and continuation. Continuation of PrEP was defined if PrEP users who come back to pick up drugs (for daily PrEP) or self-reported to adherence (for ED-PrEP) at the corresponding follow-up visits after initiation at 3-month, 6-month and 12-month visits.

### Data analysis
Participants' responses to the open-ended questions were coded by an independent investigator (VQD). Statistical analysis was performed using the IBM SPSS Statistics for Windows, V.27.0 (IBMrp). We used conventional descriptive statistics to summarise the characteristics of the study's participants and their views on PrEP. Effective use was calculated by dividing the number of PrEP users retained at 3, 6 and 12 months of PrEP by the total number of clients enrolled in PrEP study and multiplying by 100. Multivariable logistic regression was used to estimate adjusted OR (aOR) and 95% CIs for PrEP retention by selected baseline characteristics. Values of p<0.05 were considered statistically significant.

### Patient and public involvement
No patient or public involvement.

## RESULTS
Between May 2020 and April 2021, we enrolled 926 clients of whom 253 (27.3%) choose ED-PrEP and 673 (72.7%) choose daily PrEP at enrolment. Participants

**Table 1** Characteristic of participants by the initial PrEP preference

| Characteristics | All participants (n=926) | Participants preferred a daily PrEP regimen (n=673) | Participants preferred ED PrEP regimen (n=253) | P value |
|---|---|---|---|---|
| Gender identity | | | | <0.001 |
| Male | 858 (92.7%) | 608 (90.3%) | 250 (98.8%) | |
| Female | 63 (6.8%) | 61 (9.1%) | 2 (0.8%) | |
| Trans female | 5 (0.5%) | 4 (0.6%) | 1 (0.4%) | |
| Age (median, IQR) (years) | 24 (20–28) | 24 (20–28) | 23 (21–27) | |
| Sexual partners | | | | 0.568 |
| No answer | 3 (0.3%) | 3 (0.4%) | 0 (0.0%) | |
| Men exclusively | 784 (84.7%) | 569 (84.5%) | 215 (85.0%) | |
| Men and women | 139 (15%) | 101 (15%) | 38 (15%) | |
| HIV exposure within the past 3 days | | | | 0.224 |
| No HIV exposure | 915 (98.8%) | 667 (99.1%) | 248 (98.0%) | |
| HIV exposure | 10 (1.1%) | 5 (0.7%) | 5 (2.0%) | |
| No answer | 1 (0.1%) | 1 (0.1%) | 0 (0.0%) | |
| Frequency of sexual activity | | | | <0.001 |
| ≤2 times per week | 279 (32.1%) | 228 (37.0%) | 51 (20.2%) | |
| >2 times per week | 561 (64.6%) | 366 (59.4%) | 195 (77.4%) | |
| No answer | 28 (3.2%) | 22 (3.6%) | 6 (2.4%) | |
| Having sex without condom with people who were at risk of HIV within the past 6 months | | | | <0.001 |
| No | 352 (38.0%) | 183 (27.2%) | 169 (66.8%) | |
| Yes | 502 (54.2%) | 424 (63.0%) | 78 (30.8%) | |
| No answer | 72 (7.8%) | 66 (9.8%) | 6 (2.4%) | |
| No of sexual partners within the past 6 months | | | | <0.001 |
| One sexual partner | 205 (22.1%) | 171 (25.4%) | 36 (14.2%) | |
| At least two sexual partners | 703 (75.9%) | 495 (73.6%) | 208 (82.2%) | |
| No answer | 16 (1.7%) | 7 (1.0%) | 9 (3.6%) | |
| Having sex for money or gifts within the past 6 months | | | | 0.002 |
| No | 835 (90.2%) | 619 (92.0%) | 216 (85.4%) | |
| Yes | 60 (6.5%) | 32 (4.8%) | 28 (11.1%) | |
| No answer | 31 (3.3%) | 22 (3.3%) | 9 (3.6%) | |
| Diagnosis and/or treatment with an STI within the past 6 months | | | | <0.001 |
| No | 793 (85.6%) | 597 (88.7%) | 196 (77.5%) | |
| Yes | 103 (11.1%) | 51 (7.6%) | 52 (20.6%) | |
| No answer | 30 (3.2%) | 25 (3.7%) | 5 (2.0%) | |
| Sharing needles with other people within the past 6 months | | | | 0.387 |
| No | 917 (99.5%) | 664 (99.3%) | 253 (100.0%) | |
| Yes | 2 (0.2%) | 2 (0.3%) | 0 (0.0%) | |
| No answer | 3 (0.3%) | 3 (0.4%) | 0 (0.0%) | |
| Used PrEP within the past 6 months | | | | 0.001 |
| No | 874 (94.8%) | 623 (93.1%) | 251 (99.2%) | |

**Table 1** Continued

| Characteristics | All participants (n=926) | Participants preferred a daily PrEP regimen (n=673) | Participants preferred ED PrEP regimen (n=253) | P value |
|---|---|---|---|---|
| Yes | 44 (4.8%) | 42 (6.3%) | 2 (0.8%) | |
| No answer | 4 (0.4%) | 4 (0.6%) | 0 (0.0%) | |
| Willingness to pay for PrEP | | | | <0.001 |
| <US$4.25/month | 196 (21.2%) | 130 (19.3%) | 66 (26.1%) | |
| US$4.25–US$14.89/month | 511 (55.2%) | 407 (60.5%) | 104 (41.1%) | |
| US$14.89–US$42.55/month | 145 (15.7%) | 91 (13.5%) | 54 (21.3%) | |
| >US$42.55/month | 2 (0.2%) | 2 (0.3%) | 0 (0.0%) | |
| Don't want/unable to pay for PrEP | 72 (7.8%) | 43 (6.4%) | 29 (11.5%) | |

ED, event driven; PrEP, pre-exposure prophylaxis; STI, sexually transmitted infection.

were followed up until 31 December 2022. Table 1 and online supplemental table 1 show the characteristics of enrolled participants. The median age was 24 years (IQR 20–28) and ranging from 16 to 51 years. Twelve participants were under 18 years old (1.3%). The majority of men participants reported their exclusive sex with men

**Table 2** Reasons for choosing PrEP at baseline

| Reasons | All participants | Daily PrEP | ED-PrEP |
|---|---|---|---|
| No of sex partners | | | |
| Few sex partners | 25 (2.9%) | 25 (3.9%) | 0 (0.0%) |
| Multiple sex partners | 72 (8.3%) | 71 (11.1%) | 1 (0.4%) |
| Frequency of sex | | | |
| Having less frequent sex | 19 (2.2%) | 7 (1.1%) | 12 (5.2%) |
| Having more frequent sex | 104 (11.9%) | 96 (15.0%) | 8 (3.4%) |
| Effectively prevents HIV | 208 (23.9%) | 155 (24.3%) | 53 (22.7%) |
| Prevents HIV following condomless sex | 155 (17.8%) | 146 (22.9%) | 9 (3.9%) |
| Easy to adhere | 149 (17.1%) | 121 (19.0%) | 28 (12.0%) |
| Convenient | 79 (9.1%) | 37 (5.8%) | 42 (18.0%) |
| Helps protect others and oneself from HIV | 60 (6.9%) | 38 (6.0%) | 22 (9.4%) |
| Safe (no concerns about side effects) | 52 (6.0%) | 37 (5.8%) | 15 (6.4%) |
| Marriage and/or staying with spouse/partner | 33 (3.8%) | 33 (5.2%) | 0 (0.0%) |
| Having HIV positive sexual partner | 29 (3.3%) | 28 (4.4%) | 1 (0.4%) |
| Wants to try something new | 27 (3.1%) | 1 (0.2%) | 26 (11.2%) |
| Desires to feel safe (protected from HIV) | 20 (2.3%) | 20 (3.1%) | 0 (0.0%) |
| Protection during vaginal sex | 18 (2.1%) | 18 (2.8%) | 0 (0.0%) |
| Did not want to use condom | 17 (2.0%) | 8 (1.3%) | 9 (3.9%) |
| PrEP option was free of charge | 16 (1.8%) | 4 (0.6%) | 12 (5.2%) |
| Fears about broken condoms | 10 (1.1%) | 0 (0.0%) | 10 (4.3%) |
| General fears about HIV | 7 (0.8%) | 2 (0.3%) | 5 (2.1%) |
| Partner takes daily PrEP | 3 (0.3%) | 1 (0.2%) | 2 (0.9%) |
| Protects one's family | 2 (0.2%) | 2 (0.3%) | 0 (0.0%) |
| Doubts about their partner's fidelity | 2 (0.2%) | 1 (0.2%) | 1 (0.4%) |
| Protection during oral sex | 1 (0.1%) | 1 (0.2%) | 0 (0.0%) |
| Concerns about exposure to blood | 1 (0.1%) | 1 (0.2%) | 0 (0.0%) |

ED, event driven; PrEP, pre-exposure prophylaxis.

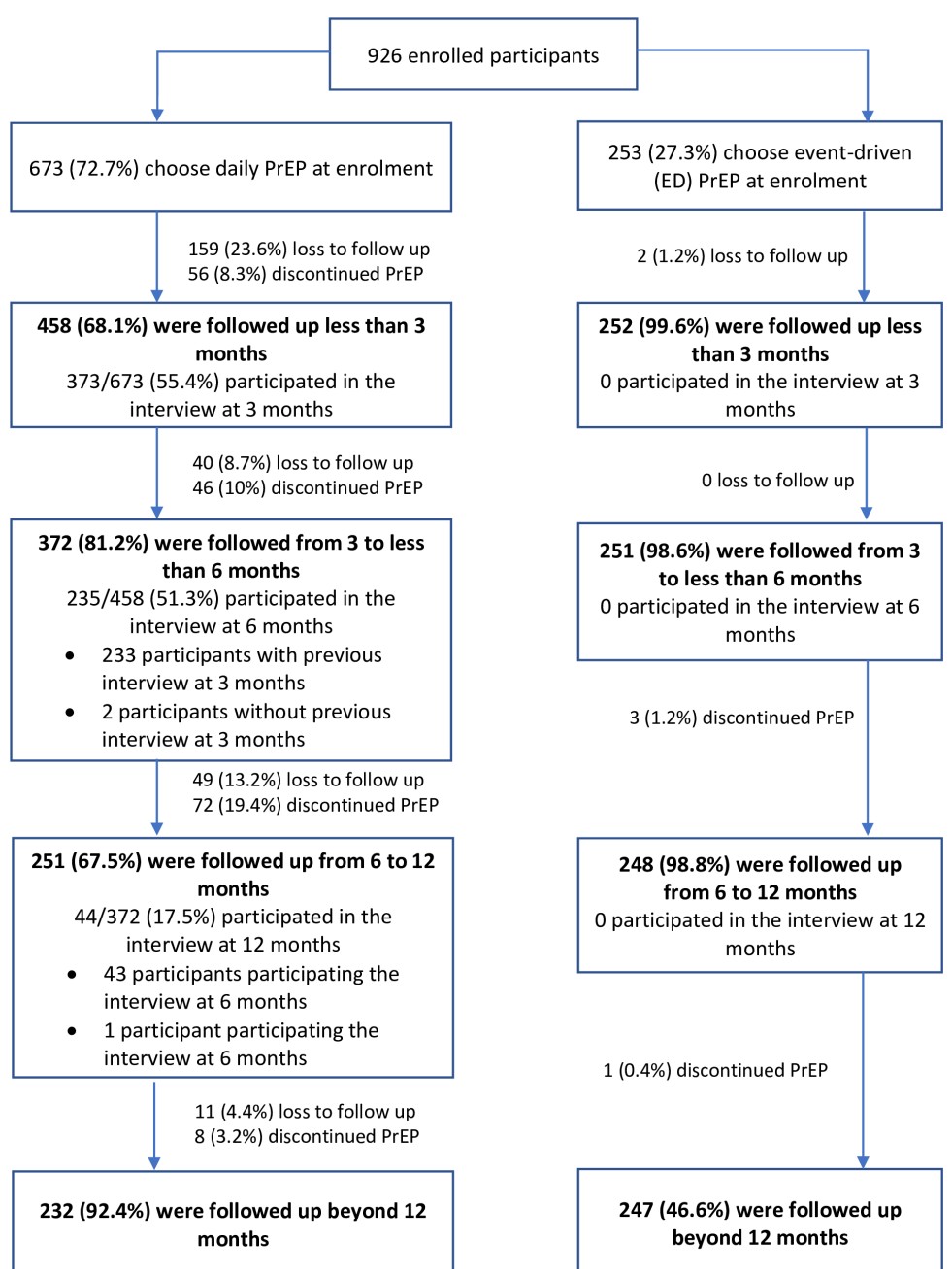

**Figure 1** Flow chart of study participants. PrEP, pre-exposure prophylaxis.

(784/926 or 84.7%) and there was no significant difference in age and gender identity between groups of daily PrEP and ED-PrEP.

At baseline, 94.1% (871/926) participants responded to an open-ended question on the reasons that determined their PrEP preference after the consultation with the clinic staff. The response rate for reporting the reasons PrEP preference was not significant between the groups of daily PrEP (94.8%; 638/673) and ED-PrEP (233/253 or 91.1%). The most common reasons for choosing daily PrEP were the effectiveness of PrEP in preventing HIV infection (155/638 or 24.3%), preventing the transmission of HIV following condomless sex (146/638 or 22.9%) and easy for longer-term use (121/638 or 19.0%). Among participants who chose ED-PrEP, the most common reasons were effectiveness (53/233 or 22.7%), convenience (42/233 or 18%) and easy for adherence (28/233 or 12.0%). The detailed reasons for PrEP preference at baseline are shown in the table 2. Among 338/926 (36.5%) participants who anticipated PrEP barriers, only 22/338 (6.5%) (17/265 preferred daily PrEP and 5/73 preferred ED-PrEP) expressed the specific concerns on PrEP (online supplemental table 2).

Regarding the willingness to pay for PrEP, most (76.4%) were willing to pay for PrEP if less than US$15 per month, while some (7.8%) said they would not pay or felt they were unable to pay for PrEP at any cost. The proportions of clients willing to pay at different prices were statistically different between daily and ED-PrEP groups (table 1).

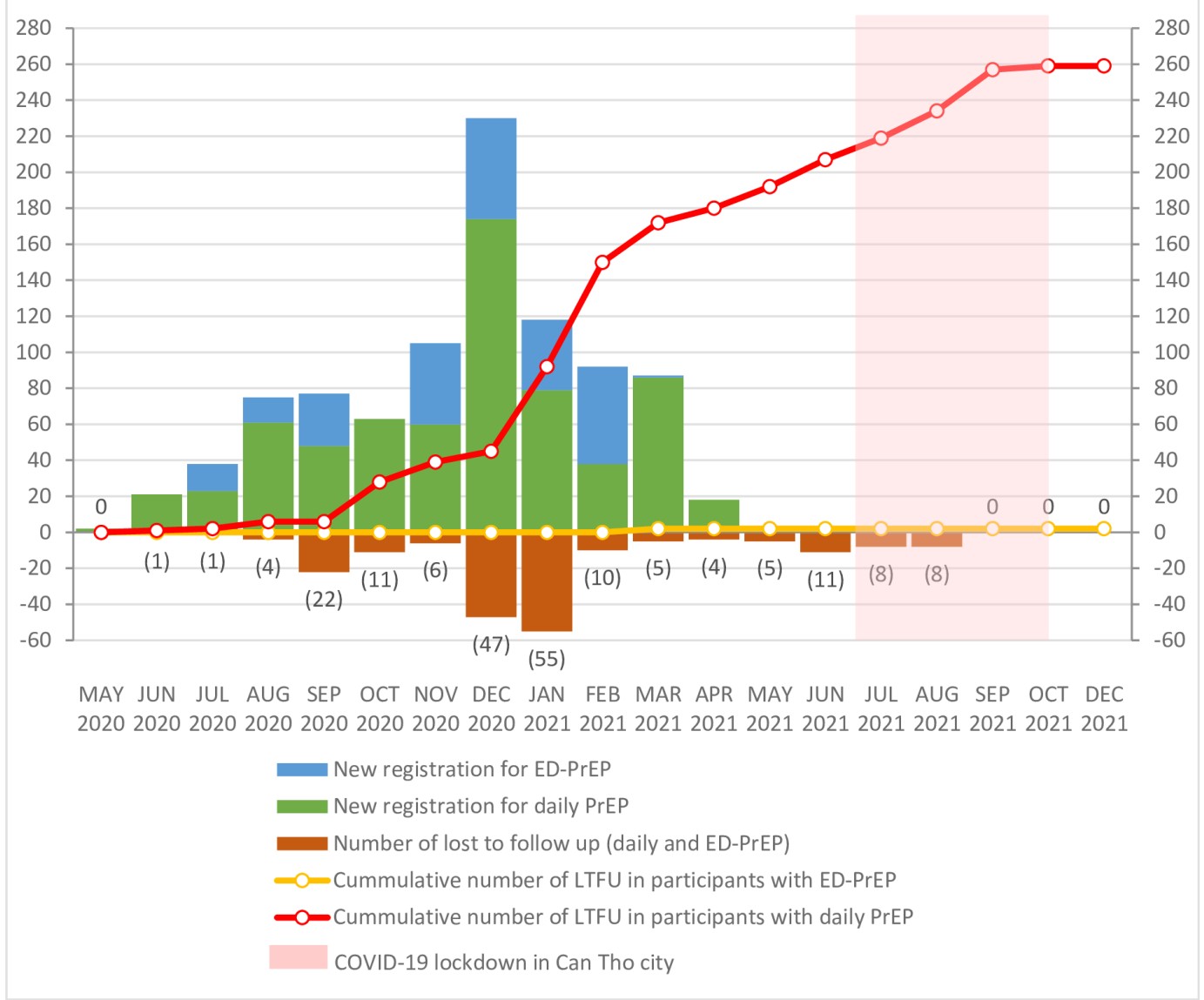

**Figure 2** Number of new PrEP initiation and lost to follow-up (LTFU) after the study enrolment. ED, event driven; PrEP, pre-exposure prophylaxis.

The median follow-up time was 284 days (IQR 102–367) among 926 participants who initiated PrEP, 214 days (IQR 60–323) in participants choosing daily PrEP and 363 days (IQR 319–389) in participants choosing ED-PrEP. By the end of the study, 261/926 (28.2%) patients were lost to follow-up, 186/926 (20.1%) discontinued and 479/926 (51.7%) were on PrEP (figure 1). Much of the loss to follow-up occurred within the first 3 months of enrolment (159/261 or 60.9%) and among those taking daily oral PrEP (259/261 or 99.2%). The overall retention rates at 3, 6 and 12 months in the daily PrEP group were 72.6% (439/605), 64.5% (363/563) and 43.1% (150/198), respectively, with the median time of lost to follow-up of 60 days. The retention rates in the ED-PrEP group were 99.2% (251/253) at 3 and 6 months and 99.4% (158/159) at 12 months. Of 186 participants who discontinued PrEP, reasons reported for discontinuation were that they were no longer sexually active (87/186 or 46.8%), moving to a new place (85/186

or 45.7%), were diagnosed with HIV (seroconversion) (7/186 or 3.8%), had different user preferences, had concerns about medication-related toxicities, were diagnosed with HBV (each of 2/185 or 1.1%) or were affected due to COVID-19-related restrictions (1/186 or 0.5%).

Of participants who started daily PrEP and completed the interview, the proportions of participants reporting any PrEP side effects were 32/341 (8.6%) at 3 months, 12/235 (5.1%) at 6 months and 5/44 (11.4%) at 12 months. The details of reported side effects are listed in online supplemental table 3.

During COVID-19 pandemic, Can Tho city was locked down due to COVID-19 outbreak (eg, from July to October 2021). However, as shown in figure 2, a larger percentage of participants discontinued PrEP before the lockdown period. There were only 40/261 (15.3%) participants who lost to follow-up within 4 months of lockdown and before of the study completion.

**Table 3** Multivariate logistic regression predicting the retention in daily PrEP in Can Tho (n=414)

| Variables | Adjusted OR (95% CI) | P value |
|---|---|---|
| Age (1-year increment) | 0.967 (0.934 to 1.002) | 0.065 |
| Gender identity | | |
| Male | 1 | |
| Female/transfemale | 1.932 (0.896 to 4.170) | 0.093 |
| Sexual partners | | |
| Men and women | 1 | |
| Men exclusively | 1.720 (0.919 to 3.218) | 0.090 |
| Frequency of sexual activity | | |
| >2 times per week | 1 | |
| ≤ 2 times per week | 2.199 (1.306 to 3.702) | **0.003** |
| No answer | 2.130 (0.927 to 4.897) | 0.075 |
| Having sex without condom with people who were at risk of HIV within the past 6 months | | |
| No | 1 | |
| Yes | 1.991 (1.180 to 3.362) | **0.010** |
| No of sexual partners within the past 6 months | | |
| ≥2 partners | 1 | |
| <2 partners | 1.225 (0.722 to 2.077) | 0.452 |
| Having sex for money or gifts within the past 6 months | | |
| Yes | 1 | |
| No | 3.259 (0.742 to 14.318) | 0.118 |
| Diagnosis and/or treatment with an STI within the past 6 months | | |
| No | 1 | |
| Yes | 0.795 (0.325 to 1.944) | 0.615 |
| Used PrEP within the past 6 months | | |
| No | 1 | |
| Yes | 13.568 (3.015 to 61.071) | **0.001** |
| Willingness to pay for PrEP | | |
| Don't want/unable to pay for PrEP | 1 | |
| Willing to pay | 0.832 (0.275 to 2.517) | 0.744 |
| Anticipated barrier to PrEP (1) | | |
| No | 1 | |
| Yes | 1.721 (1.042 to 2.842) | **0.034** |

A p-value of <0.05 was bold .
PrEP, pre-exposure prophylaxis; STI, sexually transmitted infection.

We also carried out multivariable logistic regression to determine factors associated with PrEP retention among daily PrEP participants (continued daily PrEP group vs lost-to-follow-up group). The results indicate that factors associated with daily PrEP continuation were less frequency of sex (less than twice per week) (aOR 2.199, 95% CI 1.306 to 3.702), having sex without condom with people who were at risk of HIV within the past 6 months (aOR 1.991, 95% CI 1.180 to 3.362), PrEP use within the past 6 months (aOR 13.568, 95% CI 3.015 to 61.071) and no anticipation of barriers to PrEP use (aOR 1.721, 95% CI 1.042 to 2.842) (table 3). Due to small number of lost

to follow-up among ED-PrEP users, we were unable to execute multivariable Cox regression for this group.

## DISCUSSION

This study is the first prospective study in Viet Nam to explore both PrEP preferences and use, as well as effective use and factors associated with PrEP continuation. We found that among individuals who were eligible for PrEP, more than a quarter preferred ED-PrEP over daily PrEP. The proportion of the MSM who preferred ED-PrEP in our study was similar to the studies in high-income

countries, including Belgium (23.4%–23.5%),[9 10] the Netherlands (26.7%–27.3%)[10 11] and Australia (~20%)[12 13] but was lower than countries such as France (49.5%),[14] Taiwan (56%),[15] China (57.1%)[16] and others in West Africa (72.1%–74%).[17 18] We reported a great variety of individual factors determine the choices for their PrEP regimens, mostly related to the participants' perceptions of PrEP efficacy in prevention of HIV transmission, safety, perceived adherence and convenience. In a qualitative study in 857 MSM on daily PrEP and 301 MSM on ED-PrEP in the Netherlands, preference of oral PrEP was reported to include frequency of sex, expected adherence, perceived safety, efficacy and burden of the pills and anticipated side effects.[19] In a prospective study in 1000 MSM who use oral PrEP in China, the multivariable marginal effect analysis show that factors associated with an increased preference for daily versus ED-PrEP were currently being married to or living with a female (adjusted marginal effect = −0.146 (95% CI −0.230 to −0.062), p=0.001), number of male sexual partners in the previous 6 months (adjusted marginal effect=0.003, 95% CI 0.000 to 0.005), p=0.034) and a subjective assessment of being very high risk of HIV infection (adjusted marginal effect size=0.105 (95% CI 0.012 to 0.198), p=0.027).[16]

For PrEP effective use, we found that those on ED-PrEP had greater continuation rate compared with those opting for daily oral PrEP at 12 months (99.2% and 43.1%, p<0.001, respectively). Daily PrEP retention has been reported in different studies with large variations by study design, PrEP delivery approaches and countries. The proportion of retention at 12 months ranged from 43% in a study of 5583 MSM from 2012 to 2017 in 6 clinical sites in the USA,[20] 72.3% in a cohort of 1347 PrEP users in Belgium between 2017 and 2020,[21] 83% in 450 MSM in Brazil between 2014 and 2016[22] to 91.8% in a study of 400 MSM in 12 urban US cities in 2013.[23] In a randomised control trial with 119 MSM in Hong Kong, the oral daily and ED PrEP retention at 32 weeks was 86% and 87%, respectively.[24] Proportion of retention to ED-PrEP among Thai MSM aged 15–19 years was 88.9%, 95% CI 73.9% to 96.9%) at 6 months.[25] The definition of retention for PrEP was varied and made it difficult to compare the proportion of retention across the studies. It was conventionally defined as the return for follow-up every 3 months[26] or attendance at a specific time point (eg, 3, 6 and 12 months) with a time window (±30 days) while clients may not attend a follow-up visit with a precise intervals,[27] especially in case of ED-PrEP. In addition, concepts of retention are changing and that people come during periods of risk and have different needs. Thus, effective use of PrEP is increasingly being used while adherence and retention are not. Our findings suggest more diverse and flexible PrEP models might lead to better use and engagement from clients. These results can also be leveraged to improve oral daily PrEP by making services and follow-up more differentiated including use of HIVST use for PrEP continuation. We found that the majority of MSM in Can Tho were willing to pay for PrEP (92.2%). However, 82.8% (707/854) participants in our study indicated that their willingness to pay was low (<US$15/month) (the average income per person in Mekong delta was VND3 713 000[28] or approximately US$160.3, at the exchange rate US$1=VND23 159.8 in 2021[29]) while 65% of respondents in Thailand willing to pay US$25 (monthly average income per person was US$478.1 in 2012[30]) and 88.9% of respondents in China would like to pay >US$14 per month for PrEP (monthly average income per person was US$ 867.3 in 2020[30]).

We noticed that the number of new registration for PrEP and the number of lost to follow-up were highest between December 2021 and January 2022. In Viet Nam, December was designated for the HIV action month with many events promoted for HIV interventions including PrEP, which may have been related to the increase in the new registration and also high rate of lost to follow up thereafter.

## Limitation

Our limitation in this study was the patients were recruited from a single province; thus, the study site is purposely selected and may not represent all PrEP users in Viet Nam. The self-report on challenges during PrEP taking could be recall bias. In addition, we did not include information on restarting PrEP or switching from daily PrEP to ED-PrEP in this analysis which may cause a potential bias of underestimate the retention rate. Also, self-statement of adherence among ED-PrEP users could contribute to overestimate of the retention in this group.

## CONCLUSION

Individuals at substantial risk for HIV especially MSM in Can Tho, Viet Nam were motivated to choose PrEP by their beliefs about the safety, efficacy and, frequency of sex and expected adherence with little concerns about side effects and specific barriers to use PrEP. ED-PrEP was desirable and achieved high levels of effective use in this cohort study, but with low willingness to pay. ED-PrEP is desirable and should be offered as an option to expand access and prevent new infections in Viet Nam. Further research is needed to provide more insights, particularly on loss to follow-up and implementation of more flexible PrEP delivery models.

**Acknowledgements** We are very grateful to the health staff at Can Tho CDC and the participating PrEP clinics in Can Tho who contributed to the implementation of the study.

**Contributors** VTTN and PNAT conceived and designed the study. VQD analysed the data and wrote the first draft of the report. VTTN, PNAT, HMT, PNAT and DTTL involved to the acquisition and interpretation of data. VTTN, CJ, RB and PNAT reviewed the manuscript and provided substantial contribution. VTTN is responsible for the overall content as the guarantor. All authors contributed to the manuscript finalisation and approved the final report.

**Funding** The study was funded by Viet Nam—WHO Country Office. Grant number: NA.

**Competing interests**  None declared.

**Patient and public involvement**  Patients and/or the public were not involved in the design, or conduct, or reporting, or dissemination plans of this research.

**Patient consent for publication**  Consent obtained directly from patient(s).

**Ethics approval**  This study involves human participants and the study was approved by Hanoi Medical University Institutional Ethical Review Bboard (IRB VN01.001/IRB 0000312/FWA 00004148), and WHO Wester Pacific Regional Office Ethical Review Committee (2020.4.VTN.1.HSI). Participants gave informed consent to participate in the study before taking part.

**Provenance and peer review**  Not commissioned; externally peer reviewed.

**Data availability statement**  Data are available on reasonable request.

**ORCID iDs**
Van Thi Thuy Nguyen http://orcid.org/0000-0001-8766-3082
Vu Quoc Dat http://orcid.org/0000-0002-5904-5970

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
