## [Reviewer comments · BMJ Open]

ARTICLE DETAILS

TITLE (PROVISIONAL)	Preference and retention of daily and event-driven pre-exposure prophylaxis for HIV prevention: a prospective cohort in Can Tho city, Viet Nam
AUTHORS	Dat, Vu Quoc; Nguyen, Van Thi Thuy; Truc, Huynh Minh; Nguyen Anh Thu, Pham; Linh, Doan Thi Thuy; Johnson, Cheryl; Baggaley, R; Phan, Huong

VERSION 1 – REVIEW

REVIEWER	van de Vijver, David Erasmus MC, Viroscience
REVIEW RETURNED	24-Jul-2023

GENERAL COMMENTS	The study under review is a well-written manuscript on the preference and retention of daily and event driven PrEP in Can Tho city in Viet Nam. This is an important topic as the prevalence of HIV in MSM, one of the key populations, has strongly increased in recent years. As recommended by the WHO, PrEP is a way to reduce HIV transmissions. In their paper, the authors conclude that almost 25% of participants preferred event-driven over daily PrEP. Virtually all of those preferring event-driven PrEP were retained. Conversely, more than 50% of individuals preferring daily PrEP were not retained. Most participants were willing to pay for PrEP if it cost less than 15USD per month. Major comments 1. The study seems not to have considered re-starts of PrEP in individuals that had previously discontinued. Did any participants re-start PrEP?2. The analysis included a cox proportional hazard analysis comparing the characteristics of individuals continuing daily PrEP vs individuals lost to follow up. I am concerned that Cox's assumption of proportional hazards over time is not met as those retained in care will remain constant. I therefore recommend to perform a logistic regression analysis (which most likely will have the same results). Minor comment 1. Please write out ED-PrEP in the abstract, sub-heading 'Findings' as this is the first time this abbreviation is used in the abstract.
---

REVIEWER	Chan, Curtis University of New South Wales, Faculty of Medicine
REVIEW RETURNED	18-Aug-2023

GENERAL	This is an important paper about the use of event-driven PrEP in Vietnam, and adds to
----------------	---

COMMENTS	the body of literature that there are a substantial proportion of MSM who prefer ED-PrEP and would benefit from having this option available to them. Please see my comments below. There are some significant clarifications that should be made before this work is published. Furthermore, this manuscript would benefit from additional proofreading and editing for clarity. Preference and retention of daily and event-driven pre-exposure prophylaxis: a prospective cohort in Can Tho city, Viet Nam Review Major Abstract – Findings: Change “female transgender” to “transgender women”. The term “transgenderers” has a highly negative and stigmatising connotation Method – study design and participants. Consider restricting the analysis to cisgender MSM and trans-women who are not taking gender affirming hormones, as these are the only people eligible for ED-PrEP. It is problematic there were 2 cisgender women who preferred ED-PrEP since they are not eligible. Method pg5 line 41: “ED-PrEP were offered for MSM who have infrequent sex... or their own preference of ED-PrEP”. Can you clarify if clients to this service were only informed about ED-PrEP if they were not having frequent sex? In other words, if someone was considered highly sexually active, were they only told to take daily PrEP? Based on the methods now, it is not clear whether ALL participants were given the choice, which may directly impact on one of your main findings about how many people would prefer ED-PrEP. Methods, pg 6 line 7: “Continuation of PrEP was defined if PrEP users who come back to pick up drugs (for daily PrEP) or self-reported to adherence (for ED-PrEP) at the corresponding following up visits after initiation at 3-, 6- and 12-month visits.” Based on the sentence “PrEP clients are followed up regularly at one and two months after PrEP initiation, and every 3 months thereafter”, does this mean a daily user that misses one of these appointments (1 and 2 month, then 3, 6, 9, and 12 months) would not be considered retained? ED-PrEP users only need to indicate self-reported adherence at 3, 6 and 12 months. The significant difference between retention between daily and ED-PrEP users is therefore likely to be because there are more opportunities for daily users to be lost to follow up. While there is some acknowledgement of this in the discussion section, please clarify this in your methods, and you must address this in the limitations. Minor Abstract – Strength/Limitation: “We found that event driven PrEP was preferred over daily PrEP” – this does not seem to be the case as most people (72.7%) chose daily. Abstract – keywords: consider adding “PrEP” after “daily” and “event-driven”. On their own as keywords, they may not make sense. Introduction, pg4 line 40: You have already defined ED-PrEP in line 32. Consider removing “event-driven” here. Introduction, pg 4 line 59: Typo – “PrEP r among MSM in Vietnam” Methods, pg 5 line 14: “all 11 PrEP clinics” – can you describe what is a “PrEP clinic”? Is it clinic where the only service it provides is PrEP prescribing/dispensing? Does it provide other services? Who owns/runs the clinic? Can you provide a citation that gives a description of how these clinics are run? Methods pg 5 line 45: Typo “or their own preference for ED-PrEP..” , two full stops
-----------------	--

	after ED-PrEP Methods, pg 6 line 7: “following MOH guidelines” – MOH has not been defined yet. Method/Result: Were participants able to switch between daily and ED-PrEP in the 12 months of follow up? Ethics pg 6 line 30: Typo “Review Bboard” Results pg 7 line 12: “Regarding the willingness to pay for PrEP, most (76.4%) were willing to pay for PrEP if less than \$15 per month). What currency is this in? Were participants asked in VND? If so, consider reporting it in the original currency, then were appropriate provide the estimated conversion (e.g. to USD). This would also apply to Table 1 and willingness to pay. Discussion pg 8 line 18: Some of these references for these countries are not the most up-to-date. Consider using these instead:  • Belgium/The Netherlands: Jongen et al. (2021). https://doi.org/10.1002/jia2.25768 • Australia: Chan et al. (2022). https://doi.org/10.1007/s10461-021-03344-3  ○ However, this paper is the same as Vaccher et al. (2017) as they both are about hypothetical preference rather than actual use. The ED-PrEP use among gay and bisexual male PrEP users in 2021 is about 20.5% and in 2022 was at 24.8% as presented at the IAS2023 conference – Abstract number OAC0505: https://www.iasociety.org/sites/default/files/IAS2023/abstract-book/IAS_2023_Abstracts.pdf • West Africa: Laurent et al. (2023). https://doi.org/10.1093/cid/ciad221 Also, there are some from other countries to consider  • France: Molina et al. (2022). https://doi.org/10.1016/S2352-3018(22)00133-3 • Germany: Koppe et al. (2021). https://doi.org/10.1186/s12889-021-10174-4 Discussion pg 8 line 52: Please provide a citation for the Hong Kong MSM study. I believe you want to cite Kwan et al. (2021) JIAS. Table 2: Many of the reasons are too vague. E.g. it is unclear how many is ‘few’ and ‘multiple’. It is unclear what is meant by ‘Unplanning for sex’, ‘Self-esteem’, ‘Match the personal risk’, ‘New programme’, ‘Check for body tolerance’. Also, it is not clear why ‘Want to have a child’ would affect their PrEP choice. Figure 1: Typo: “beyond 12 months” - beyond Discretionary/Optional Throughout: where relevant, consider adding “oral PrEP” – this will clarify that you are not discussing CAB-LA or the dapivirine ring. E.g. “data on oral PrEP preference” in the objectives Throughout: you switch between “Vietnam” and “Viet Nam”, as well as “event-driven” and “event driven”. Consider choosing one for consistency. Method, pg 5 line 29: Consider changing “sexually transmitted disease (STD)” to “sexually transmitted infection (STI)”.
--	--

Reviewer 1: Dr. David van de Vijver, Erasmus MC

Comments to the Author:

The study under review is a well-written manuscript on the preference and retention of daily and event driven PrEP in Can Tho city in Viet Nam. This is an important topic as the prevalence of HIV in MSM, one of the key populations, has strongly increased in recent years. As recommended by the WHO, PrEP is a way to reduce HIV transmissions. In their paper, the authors conclude that almost 25% of participants preferred event-driven over daily PrEP. Virtually all of those preferring event-driven PrEP were retained. Conversely, more than 50% of individuals preferring daily PrEP were not retained. Most participants were willing to pay for PrEP if it cost less than 15USD per month.

Major comments

1. The study seems not to have considered re-starts of PrEP in individuals that had previously discontinued. Did any participants re-start PrEP?

Our response: Thank you very much for your important comment. Indeed, we did not have information of patients who restarted PrEP. We recognized this is a potential bias that has been discussed in the limitation as below:

“In addition, we didn't included information on patients who re-started PrEP or switched from daily PrEP to ED-PrEP in this analysis which may cause a potential bias of underestimate the retention rate, especially in PrEP users”.

2. The analysis included a cox proportional hazard analysis comparing the characteristics of individuals continuing daily PrEP vs individuals lost to follow up. I am concerned that Cox's assumption of proportional hazards over time is not met as those retained in care will remain constant. I therefore recommend to perform a logistic regression analysis (which most likely will have the same results).

Our response: Thank you for this valuable comment. We agreed that the logistic regression analysis seems more appropriate. We have re-analysed the data and presented in the revised table as below.

Variables	Adjusted odd ratio (95%CI)	P value
Age (1-yr. increment)	0.967 (0.934-1.002)	0.065
Gender identity		
Male	1	
Female/trans female	1.932 (0.896-4.170)	0.093
Sexual partners		
Men and women	1	
Men exclusively	1.720 (0.919-3.218)	0.090
Frequency of sexual activity		
>2 times per week	1	
≤ 2 times per week	2.199 (1.306-3.702)	0.003
No answer	2.130 (0.927-4.897)	0.075
Having sex without condom with people who were at risk of HIV within the past 6 months		
No	1	
Yes	1.991 (1.180-3.362)	0.010
Number of sexual partners within the past 6 months		
≥2 partners	1	
<2 partners	1.225 (0.722-2.077)	0.452
Having sex for money or gifts within the past 6 months		

Yes 1
No 3.259 (0.742-14.318) 0.118
Diagnosis and/or treatment with an STI within the past 6 months
No 1
Yes 0.795 (0.325-1.944) 0.615
Used PrEP within the past 6 months
No 1
Yes 13.568 (3.015-61.071) 0.001
Willingness to pay for PrEP
Don't want/unable to pay for PrEP 1
Willing to pay 0.832 (0.275-2.517) 0.744
Anticipated barrier to PrEP (1)
No 1
Yes 1.721 (1.042-2.842) 0.034

Minor comment

1. Please write out ED-PrEP in the abstract, sub-heading 'Findings' as this is the first time this abbreviation is used in the abstract.

Our response: Thank you! We have added the full spelling for abbreviation of ED-PrEP (event-driven PrEP) when we first use.

Reviewer 2. Dr. Curtis Chan, University of New South Wales

Comments to the Author:

This is an important paper about the use of event-driven PrEP in Vietnam, and adds to the body of literature that there are a substantial proportion of MSM who prefer ED-PrEP and would benefit from having this option available to them.

Please see my comments in the attached word file. There are some significant clarifications that should be made before this work is published. Furthermore, this manuscript would benefit from additional proofreading and editing for clarity.

Major

Abstract – Findings: Change “female transgender” to “transgender women”. The term “transgender” has a highly negative and stigmatising connotation.

Thank you for your advice. We have changed the term as per your advice

Method – study design and participants. Consider restricting the analysis to cisgender MSM and trans-women who are not taking gender affirming hormones, as these are the only people eligible for ED-PrEP. It is problematic there were 2 cisgender women who preferred ED-PrEP since they are not eligible.

Our response: Thank you for your point. In our study, there were 63 MSM (including 2 those preferred ED-PrEP) who identified themselves as female. These participants are considered as transgender women. We have revised this in the main text.

Method pg5 line 41: “ED-PrEP were offered for MSM who have infrequent sex... or their own preference of ED-PrEP”. Can you clarify if clients to this service were only informed about ED-PrEP if they were not having frequent sex? In other words, if someone was considered highly sexually active, were they only told to take daily PrEP? Based on the methods now, it is not clear whether ALL participants were given the choice, which may directly impact on one of your main findings about how many people would prefer ED-PrEP.

Our response: Thank you for your comment. At the time of this study conducted in 2019-2020, the national recommendations for ED-PrEP is following WHO previous recommendations which is MSM with less frequency of sex (less than 2 per week) was eligible for ED-PrEP. We are now supporting MOH to revise the national guidelines to include all sexual active clients for ED-PrEP

Methods, pg 6 line 7: “Continuation of PrEP was defined if PrEP users who come back to pick up drugs (for daily PrEP) or self-reported to adherence (for ED-PrEP) at the corresponding following up visits after initiation at 3-, 6- and 12-month visits.” Based on the sentence “PrEP clients are followed up regularly at one and two months after PrEP initiation, and every 3 months thereafter”, does this mean a daily user that misses one of these appointments (1 and 2 month, then 3, 6, 9, and 12 months) would not be considered retained? ED-PrEP users only need to indicate self-reported adherence at 3, 6 and 12 months. The significant difference between retention between daily and ED-PrEP users is therefore likely to be because there are more opportunities for daily users to be lost to follow up. While there is some acknowledgement of this in the discussion section, please clarify this in your methods, and you must address this in the limitations.

Our response : Thank you very much for your important comment. The participants who were unable to come to the clinic on the appointment date but received their medicines through their peer or postage, for example, were considered as in retention. For the retention of ED-PrEP, yes, we totally agreed that self-statement of adherence among ED-PrEP users could contribute to overestimate of the retention in this group. We have added this point in the limitations.

Minor

Abstract – Strength/Limitation: “We found that event driven PrEP was preferred over daily PrEP” – this does not seem to be the case as most people (72.7%) chose daily.

Our response Thank you for pointing this out. We have removed this sentence.

Abstract – keywords: consider adding “PrEP” after “daily” and “event-driven”. On their own as keywords, they may not make sense.

Our response : Thank you, we have revised as per your advice.

Introduction, pg4 line 40: You have already defined ED-PrEP in line 32. Consider removing “eventdriven” here.

Our response: Thank you, we have removed it.

Introduction, pg 4 line 59: Typo – “PrEP r among MSM in Vietnam”

Our response: Thank you! We have fixed the typo.

Methods, pg 5 line 14: “all 11 PrEP clinics” – can you describe what is a “PrEP clinic”? Is it clinic where the only service it provides is PrEP prescribing/dispensing? Does it provide other services? Who owns/runs the clinic? Can you provide a citation that gives a description of how these clinics are run?

Our response: Thank you for your question. The term of PrEP clinic in this study means a public or private clinic that provide PrEP services. In Viet Nam there is no standalone PrEP clinic but rather integrated with HIV testing and/or ART clinic. We have added one sentence on clarification of PrEP clinic in the main text: All PrEP clinics were integrated with HIV testing and/or ART services

Methods pg 5 line 45: Typo “or their own preference for ED-PrEP..” , two full stops after ED-PrEP

Methods, pg 6 line 7: “following MOH guidelines” – MOH has not been defined yet.

Our response: Thank you for your suggestions we have fixed these.

Method/Result: Were participants able to switch between daily and ED-PrEP in the 12 months of follow up?

Our response: Yes, participants could switch from daily PrEP to ED-PrEP if they are eligible and preferable. We have added this information in the appendix as below:

Table supplementary. Number of participants switching between daily and ED-PrEP during the following up period

Regimen Participants preferred a daily PrEP regimen
During the first 3 months of PrEP use Daily PrEP 296 80.0%
ED-PrEP 57 15.4%
Switching between daily and ED-PrEP 17 4.6%
Subtotal 370 100.0%
Between 3 months and 6 months of PrEP use Daily PrEP 190 81.2%
ED-PrEP 41 17.5%
Switching between daily and ED-PrEP 3 1.3%
Subtotal 234 100.0%
Between 9 and 12 months of PrEP use Daily PrEP 36 83.7%
ED-PrEP 7 16.3%
Switching between daily and ED-PrEP 0 0.0%
Subtotal 43 100.0%

Ethics pg 6 line 30: Typo "Review Bboard"

Our response: Thank you. We have fixed the typo.

Results pg 7 line 12: "Regarding the willingness to pay for PrEP, most (76.4%) were willing to pay for PrEP if less than \$15 per month). What currency is this in? Were participants asked in VND? If so, consider reporting it in the original currency, then were appropriate provide the estimated conversion (e.g. to USD). This would also apply to Table 1 and willingness to pay.

Our response: Thank you for your comment. We have revised \$ to US\$ for clarity and revised the Method as below:

"We used a questionnaire consisting of six questions on willingness to pay (closed-end questions and willingness to pay estimates were reported in 2,000 Viet Nam Dong (VND) during the interview and converted to US dollar based on the official exchange rate of 2021 for this analysis (1 US\$ = 23,529 Viet Nam Dong)"

Discussion pg 8 line 18: Some of these references for these countries are not the most up-to-date. Consider using these instead:

- Belgium/The Netherlands: Jongen et al. (2021). <https://doi.org/10.1002/jia2.25768>
 - Australia: Chan et al. (2022). <https://doi.org/10.1007/s10461-021-03344-3>
o However, this paper is the same as Vaccher et al. (2017) as they both are about hypothetical preference rather than actual use. The ED-PrEP use among gay and bisexual male PrEP users in 2021 is about 20.5% and in 2022 was at 24.8% as presented at the IAS2023 conference – Abstract number OAC0505: https://www.iasociety.org/sites/default/files/IAS2023/abstractbook/IAS_2023__Abstracts.pdf
 - West Africa: Laurent et al. (2023). <https://doi.org/10.1093/cid/ciad221>
- Also, there are some from other countries to consider

- France: Molina et al. (2022). [https://doi.org/10.1016/S2352-3018\(22\)00133-3](https://doi.org/10.1016/S2352-3018(22)00133-3)
- Germany: Koppe et al. (2021). <https://doi.org/10.1186/s12889-021-10174-4>

Discussion pg 8 line 52: Please provide a citation for the Hong Kong MSM study. I believe you want to cite Kwan et al. (2021) JIAS.

Thank you very much for providing us these useful references. We have added the suggested references in our discussion section.

Table 2: Many of the reasons are too vague. E.g. it is unclear how many is 'few' and 'multiple'. It is unclear what is meant by 'Unplanning for sex', 'Self-esteem', 'Match the personal risk', 'New programme', 'Check for body tolerance'. Also, it is not clear why 'Want to have a child' would affect their PrEP choice.

Thank you for your comment. Since this is an open-ended question, the participants shared their actual thought which may be not entirely correct. There are also some issues with translation. We have revised and remove irrelevant responses to avoid confusion.

Figure 1: Typo: "beyone 12 months" – beyond
Our response: Thank you, typo was fixed.

Discretionary/Optional

Throughout: where relevant, consider adding "oral PrEP" – this will clarify that you are not discussing CAB-LA or the dapivirine ring. E.g. "data on oral PrEP preference" in the objectives

Our response: Thank you! We have revised the text as advised.

Throughout: you switch between "Vietnam" and "Viet Nam", as well as "event-driven" and "event driven". Consider choosing one for consistency.

Our response: We have revised the text as advised for its consistency.

Method, pg 5 line 29: Consider changing "sexually transmitted disease (STD)" to "sexually transmitted infection (STI)".

Our response: We have revised the text as advised.

VERSION 2 – REVIEW

REVIEWER	van de Vijver, David Erasmus MC, Viroscience
REVIEW RETURNED	19-Dec-2023
GENERAL COMMENTS	Thank you for addressing my comments.
REVIEWER	Chan, Curtis University of New South Wales, Faculty of Medicine
REVIEW RETURNED	06-Dec-2023

GENERAL COMMENTS	You have adequately addressed our comments. However, Table 2. Reasons for choosing PrEP at baseline, remains unchanged despite noting issues with translation and intending to remove irrelevant responses. At minimum, the translations should be corrected to make each reason clear/grammatically correct (e.g. "unplanning for sex", "easy to remember for taking PrEP or adherence", "no specify", and other reasons are not grammatically correct).
---

VERSION 2 – AUTHOR RESPONSE

Reviewer: 2

Dr. Curtis Chan, University of New South Wales

Comments to the Author:

You have adequately addressed our comments. However, Table 2. Reasons for choosing PrEP at baseline, remains unchanged despite noting issues with translation and intending to remove irrelevant responses. At minimum, the translations should be corrected to make each reason clear/grammatically correct (e.g. "unplanning for sex", "easy to remember for taking PrEP or adherence", "no specify", and other reasons are not grammatically correct).

Our response: Thank you for your comment, it was our error to miss the translation correction in the manuscript, we have revised the table 2 as in the attachment.